# Enhancing Rural Healthcare Accessibility: A Model for Pharmacogenomics Adoption via an Outreach-Focused Integration Strategy

**DOI:** 10.3390/jpm15030110

**Published:** 2025-03-13

**Authors:** Jared Silver, Evan Forman, David Barrett, Jovana Sibalija, Richard Kim

**Affiliations:** 1Ivey Business School, University of Western Ontario, London, ON N6G 0N1, Canada; jsilver.hba2024@ivey.ca (J.S.); eforman.hba2025@ivey.ca (E.F.); dbarrett@ivey.ca (D.B.); 2Faculty of Health Sciences, University of Western Ontario, London, ON N6A 3K7, Canada; jsibalija@ivey.ca

**Keywords:** pharmacogenomics, rural healthcare, personalized medicine, precision medicine, telemedicine, healthcare accessibility, outreach strategies

## Abstract

**Background/Objectives**: Pharmacogenomics is an emerging field in precision medicine that aims to improve patient outcomes by tailoring drug selection and dosage to an individual’s genetic makeup. However, patients in rural communities often cannot take advantage of specialized services such as pharmacogenomics due to various barriers that limit access to healthcare. This article aims to identify the barriers to implementing pharmacogenomic initiatives in rural communities and assess strategies for integrating pharmacogenomics into rural healthcare systems. **Methods**: This article describes the qualitative research that was conducted using semi-structured interviews with various stakeholders in addition to explaining how strategic frameworks were used to synthesize secondary research. **Results**: The findings of this article indicated mixed awareness of pharmacogenomics as an option amongst stakeholders, highlighting the need for targeted outreach and education intervention. Solutions such as mail-in testing and telemedicine were determined to be feasible solutions to address various geographical and logistical barriers that exist for rural patients. This article determines that successful strategies will leverage existing infrastructure and prioritize patient care, workflow integration, and adoption. **Conclusions**: Making pharmacogenomics a viable option for rural patients will take a multi-faceted approach that combines outreach, education, and innovative delivery models to overcome the multiple barriers facing rural communities.

## 1. Introduction

Pharmacogenomics (PGx) is an emerging field in precision medicine that aims to improve patient outcomes by tailoring drug selection and dosage to an individual’s genetic makeup [1]. By optimizing drug efficacy and minimizing adverse drug reactions (ADRs), PGx has the potential to enhance patient safety, reduce hospitalizations, and lower healthcare costs [2,3,4]. Despite Canada’s commitment to universal healthcare, rural populations continue to face significant barriers in accessing high-quality care compared to their urban counterparts. Rural healthcare systems are often understaffed and under-resourced, limiting access to specialized services such as PGx [5]. These healthcare disparities are particularly pronounced when it comes to emerging fields like PGx [6]. Rural communities lack not only the technological infrastructure but also the specialized workforce needed to provide precision medicine services [7,8]. Moreover, healthcare providers in rural areas often face difficulties in adopting new technologies, including PGx, due to a lack of training and the infrastructural challenges inherent in these communities [8]. Despite Canada’s commitment to universal healthcare, rural populations continue to face significant barriers in accessing high-quality care compared to their urban counterparts.

Rural healthcare systems are often understaffed, have limited access to specialized medical services, and have geographic barriers that make in-person consultations and specialized services extra challenging [5,6,7,8]. Moreover, rural healthcare providers struggle to adopt PGx due to limited training and infrastructure. While telemedicine and other digital health innovations have improved rural healthcare delivery, the integration of PGx into these systems remains underexplored. Previous studies have acknowledged these disparities but have not provided a structured, outreach-focused framework for overcoming them.

This study addresses this gap by investigating the barriers and opportunities for PGx adoption in rural healthcare settings and proposing a targeted outreach and implementation strategy. By utilizing approaches such as telemedicine and mail-in testing, it is possible to make PGx both accessible and sustainable in these underserved areas [9,10]. Furthermore, the study will identify the key factors driving the adoption of PGx among rural healthcare providers and patients, with a particular emphasis on education, accessibility, and trust in new medical technologies.

Ultimately, this study aims to identify the barriers to introducing PGx initiatives in rural communities, as well as assess various options to determine the most valuable execution method.

## 2. Materials and Methods

This study aimed to identify key barriers, opportunities, and feasibility factors for implementing PGx in rural healthcare settings. A multi-method qualitative approach was employed, involving semi-structured interviews with eight key stakeholder respondents and a qualitative review of secondary literature focused on rural healthcare and PGx.

Stakeholder respondents were selected using a purposive sampling method to ensure diverse representation across different roles in healthcare. Appendix A shows a breakdown of the sample interviewed by role. To gain comprehensive insights, semi-structured interviews were conducted with eight participants across the healthcare landscape. General Practitioners (GPs) were interviewed to understand both the clinical feasibility of PGx and the most effective ways to communicate its benefits in order to encourage patient referrals. Internal Medicine Residents provided insights into both the existing knowledge level of PGx within their training and the structure of potential continued education in this area. Pharmacists and Pharmacy Owners were interviewed to assess the operational challenges and possible roles they might play in delivering PGx services. Conversations with Patient Partners helped to gauge patient attitudes, accessibility concerns, and dynamics in patient–provider relationships specific to rural settings.

The semi-structured interview guides were developed based on key themes identified in the literature on PGx adoption and rural healthcare challenges, with a focus on understanding provider perspectives, patient accessibility concerns, and implementation feasibility.

Initial drafts of the interview guides were created to ensure alignment with the study’s core research objectives. Given the diverse range of stakeholders (General Practitioners, Internal Medicine Residents, Pharmacists, and Patient Partners), the questions were tailored to reflect the unique experiences and decision-making processes of each group.

The guides were refined through internal discussions among the research team to improve clarity, logical flow, and relevance to the study’s aims. Adjustments were made to ensure open-ended questions allowed for flexibility while maintaining consistency across interviews.

While no formal pilot study was conducted, feedback was gathered during early interviews, allowing for minor refinements in question phrasing and emphasis to enhance clarity and engagement. The final versions of the interview guides are available in Appendix A.

Secondary information was collected through a qualitative review of relevant literature on rural healthcare challenges, PGx, and healthcare advancements. A content analysis was used to identify common sentiments surrounding rural healthcare as well as how healthcare innovations can assist in the interlocution of PGx in these communities. This thematic synthesis allowed the study to establish a contextual foundation and provided insight into key trends and policy issues.

To analyze the findings and evaluate the feasibility of PGx integration, multiple strategic frameworks were applied to assess alignment between the project’s goals and the realities of rural healthcare. Frameworks such as the Diamond-E (Appendix A) and Porter’s Five Forces (Appendix A) were used to identify internal and external alignment factors, evaluating aspects like organizational resources, environmental conditions, stakeholder roles, and strategic drivers in rural contexts [11,12]. Each framework supported a layered understanding of the challenges and pathways for effective PGx outreach and implementation. Both frameworks are appropriate for the research context as they are often employed to assist with identifying gaps in a market and developing a well-rounded business plan, each of which are crucial for successfully implementing PGx initiatives (Figure 1).

Additionally, a decision matrix was employed to prioritize strategic initiatives based on criteria such as accessibility, feasibility, scalability, cost-effectiveness, and community acceptance. This matrix enabled an evidence-based approach to selecting initiatives with the greatest potential impact for rural Ontario. The Decision Matrix Method (DMM) is a tool used when evaluating alternatives actions [13,14]. The process of choosing alternatives using DMM consists of 4 key steps: Evaluation Criteria Generation, Criteria List Refinement, Relative Weights Assignments, and finally, Evaluating Alternatives against the Criteria. For this study, each criteria have the same weight. Each one is equally crucial to executing a PGx initiative in rural communities.

Based on insights from the stakeholder interviews, literature synthesis, and framework analysis, a step-by-step outreach and implementation plan was developed. This plan outlined targeted strategies for rural healthcare settings, incorporating service models like mail-in testing and telemedicine consultations to improve accessibility and foster adoption of PG.

## 3. Results

### 3.1. Overview of Rural Healthcare Challenges and Literature Review

Most government intervention is short-term oriented, rather than composing long-term, sustainable solutions [15]. Policy decisions are often guided by urban healthcare models, without fully understanding the adaptations needed to provide an efficient and equitable solution in rural communities. Recent healthcare literature shows that access to primary and specialized healthcare services is a significant issue in rural areas, where hospitals often serve as primary care providers due to the scarcity of GP services [16]. This results in high hospitalization rates and an overwhelmed healthcare system, with rural residents needing to travel longer distances for care. Additionally, a 2011 BMC Health Services study found that people in the most rural areas were less likely to have had a flu shot, use specialized services, or have a regular doctor, unlike those in urban areas [17].

Urban areas comprise most of the population in Ontario; however, their medical needs differ significantly from those in secluded areas. Public health units (PHUs) deliver programs to reduce preventable diseases and promote community health. In 2019, the Ontario government reduced the number of PHUs in the province from 35 to 10, decreasing cost burdens but ignoring unique community healthcare trends. This reduction diminishes the visibility of under-represented areas, worsening rural health outcomes.

Rural healthcare facilities also often face technological limitations, lacking the advanced equipment necessary for specialized testing [18,19]. Internet connectivity is a critical factor for telemedicine services. A 2021 report showed that 59.5% of rural/remote households in Canada have minimum connection speeds, compared to 99.3% in urban areas. In Ontario, 99.2% of urban households and 57.1% of rural/remote households have access to these speeds. However, 96.3% of rural/remote households in Canada, including 98.55% in Ontario, have access to LTE mobile cellular technology [19].

### 3.2. Organizational Insights and Strategic Fit

Frameworks such as the Diamond-E, Porter’s 5 forces, as well as the Strategy Triangle have been leveraged in order to identify key barriers and opportunities to expand PGx to rural communities. Ultimately, the findings can be used to ensure the chosen strategy allows for long-term growth and takes advantage of available resources.

#### 3.2.1. Economic Analysis

The overall strategy chosen to execute a PGx initiative into rural settings must consider economic feasibility. The strategy must not be too costly to implement and take into account regular operational expenses. The study aims to not only reduce implementation costs but also costs for patients receiving PGx testing.

Currently, collecting samples in rural areas for PGx testing involves logistical and financial challenges. LifeLabs offers a home visit service, which is currently priced at $85 per visit designed for patients with health, mobility, or geographic limitations. However, this service has coverage gaps; patients farther from a LifeLabs facility are less likely to qualify for mobile visits [20].

For patients not covered by home visit services, traveling to healthcare facilities introduces additional costs. Local transportation services can help mitigate these costs; however, are not usually prevalent in rural communities. Costs for such services in rural Ontario, for example, can cost upwards of $10 round trip, to a healthcare facility and back [21,22].

Mobile clinics are another option but require significant resources, including vehicle costs, maintenance, staffing, and community outreach, making them impractical for anything other than a large-scale, pre-emptive testing campaign. Depending on the purpose of the mobile clinic, operating costs may range anywhere from $300,000 to $2,000,000 per year [23].

When it comes to patients mailing in samples there are also significant costs relating to the transportation from the point of collection to the testing center. Couriers have various price points, which take urgency as well as distance into account, depending on the patient and their location.

Finally, a systematic review evaluating the cost-effectiveness of pharmacogenetic–guided treatment found that 71% of studies determined testing was cost-effective or cost-saving [24]. Another review concluded that most of the PGx-guided treatments that were evaluated were cost-effective compared with standard treatment as the cost of PGx testing continued to decline [25].

#### 3.2.2. Competitive Analysis

The PGx market is growing rapidly, with a projected compound annual growth rate (CAGR) of 8.52% from 2023 to 2030. However, some services do exist that feature mail-in kits and industry education.

The traditional method of prescribing medication without genetic testing serves as an alternative to PGx. While PGx offers more personalized and potentially effective treatment, the cost, accessibility, and patient acceptance of traditional methods could pose a threat, especially in cases where time is of the essence. While PGx offers more personalized and potentially effective treatment, the cost, accessibility, and patient acceptance of traditional methods remain relevant factors.

### 3.3. Stakeholder Perspectives (Appendix A)

To successfully implement PGx in rural healthcare settings, it is essential to consider the insights and perspectives of key stakeholders directly involved in patient care and service delivery. Understanding these perspectives provides practical insights into potential barriers, facilitators, and the support required for effective implementation. This study gathered qualitative feedback through semi-structured interviews to evaluate the feasibility and acceptance of PGx in these communities.

#### 3.3.1. General Practitioners and Internal Medicine Residents (Appendix A)

Both GPs and internal medicine residents expressed optimism about PGx’ potential to improve patient care through reduced adverse drug reactions and more precise medication management. While internal medicine residents tended to have slightly greater familiarity with PGx due to recent academic exposure, both groups shared key insights that could enhance adoption and referral rates.

A primary driver for adoption among both groups is the availability of robust data demonstrating PGx’ efficacy in improving patient outcomes. GPs and residents alike emphasized that seeing tangible evidence of benefit, particularly in reducing medication-related issues, would increase their confidence in referring patients for PGx testing. Clear case studies or pilot results could play a crucial role in demonstrating PGx’s value and establishing trust.

“If somebody can clearly articulate the study and help me understand the what the study was and what the outcomes were, that will really convince me quickly.” [26]

Both groups also underscored the need for seamless integration into existing workflows. To facilitate referrals, they suggested that PGx processes be accessible and ideally automated within current practice management systems. A streamlined, low-friction approach would minimize the administrative burden on practitioners, making them more inclined to incorporate PGx into their routines.

Additionally, maintaining an active role in the PGx care of their patients was seen as crucial. Both GPs and residents indicated that staying informed about their patients’ PGx results and the resulting treatment plans would make them more likely to refer future cases. They emphasized that regular updates or summaries on patient outcomes related to PGx would help them gauge its ongoing value.

Both groups expressed interest in comprehensive educational materials tailored to help them understand the referral process thoroughly. They recommended materials that clearly outline how to refer patients to the lab, identify appropriate cases for referral, and explain the process in a way that can be communicated effectively to patients. Such resources would empower them to discuss PGx confidently with their patients, ultimately increasing referral likelihood.

#### 3.3.2. Pharmacists

Pharmacists acknowledged the potential of PGx to improve medication safety and efficacy by tailoring drug prescriptions to individual genetic profiles. While they do not require detailed knowledge to interpret PGx results directly, they recognized the value of having sufficient understanding of PGx benefits to confidently educate patients. Pharmacists were confident that if they could increase patient awareness of pharmacogenomic benefits there would very likely be an increase in adoption, a theme consistent with the current literature [27].

Quote: “People are really worried about the side effects of medications. If we could explain to people that by getting this test, they wouldn’t have to deal with things like nausea, I think they would be all over it.”[28]

Additionally, pharmacists highlighted the importance of understanding how PGx services could integrate into their current billing model. They sought clarity on the financial implications, such as reimbursement options or additional billing codes that would compensate for their time spent in patient education and referrals related to PGx. They viewed this potential service as an extension of their patient counseling role, provided it aligned with existing financial structures to ensure practical integration into their daily operations.

#### 3.3.3. Patient Partners (Appendix A)

Patient partners highlighted the importance of delivering high-quality, empathetic care remotely, given the geographical and logistical challenges faced by rural residents. One key concern was ensuring that patients feel valued and understood during virtual consultations, with particular emphasis on the need for healthcare providers to establish a personal connection through telemedicine. Patient partners expressed that video consultations could sometimes lack the warmth and attentiveness of in-person visits, potentially impacting patient trust and engagement. They suggested training for providers on how to convey empathy and attentiveness through virtual platforms, enabling patients to feel supported and confident in their care.

“Each person has their own story. Each person has their own problems. [Doctors] cannot be expected to know those problems. [Doctors] cannot be expected to get into the touchy-feely [topics]. But [for proper patient care] they have to.”[29]

Additionally, patient partners emphasized the necessity of providing clear, accessible instructions for completing saliva-based test kits used in PGx. They suggested offering instructions in multiple formats, such as written, video, and visual aids, to accommodate different learning preferences and literacy levels. Language diversity was also noted as essential, with a recommendation for instructions in various languages to ensure inclusivity.

Patient partners also advocated for a structured support system at local pharmacies, allowing patients who need assistance with the testing process to receive guidance from pharmacists. This approach was seen as beneficial in helping patients complete their tests accurately, thereby reducing potential barriers to accessing PGx testing.

Finally, patient partners stressed the importance of using non-technical language when explaining PGx and its benefits. They highlighted that patients need a clear, relatable explanation of how PGx could improve their health outcomes, ideally presented in a way that empowers them to make informed healthcare decisions. Simplifying complex concepts and framing PGx in terms of tangible patient benefits would be key to enhancing patient understanding and acceptance of the testing.

### 3.4. Feasibility Analysis and Strategic Priorizitation (Decision Matrix)

The decision matrix analyzes potential methods for expanding PGx testing to rural Ontario. The analysis considers the internal, external, and primary research in order to create a comprehensive and realistic plan. Criteria includes feasibility, accessibility, cost-effectiveness, community acceptance, scalability, implementation time, and sustainability. Each alternative is rated on a scale of 1–3 for how successfully they meet each decision criteria (3 meeting criteria needs better than 1) before being given a final score out of 21. Ratings were given based on comprehensive research of each option, as well as potential barriers for execution, including economic and technological factors. The alternatives chosen include a variety of methods commonly used for spreading awareness and increasing accessibility for medical products and services (Figure 2).

**Note:** Scoring is based on a scale of 1–3, with 3 indicating the highest potential in each criterion.

The highest ranking strategies included:Mail-in Testing Kits (Score: 21): Addressed accessibility and patient convenience, minimizing travel barriers.Telemedicine Services (Score: 19): Enabled remote consultations, ensuring continuity of care in rural areas.Pharmacy-Led Initiatives (Score: 18): Positioned pharmacists as educators and patient liaisons, enhancing PGx awareness.Utilizing Local Libraries (Score: 19): Served as community education hubs for PGx literacy.

Lower-scoring options, such as mobile testing units and transportation partnerships, received lower feasibility scores due to financial and logistical constraints.

## 4. Discussion

The study’s findings highlight key insights into the challenges and opportunities for PGx adoption in rural areas and provided a foundation for a practical, targeted action plan. Through conversations with various stakeholders as well as literature analysis, it became evident that the successful integration of PGx into rural healthcare would rely heavily on addressing two primary areas: logistical accessibility and healthcare partner engagement.

Overall, the lowest ranking options tended to partially score poorly in cost-effectiveness, and scalability criteria. They each have clear benefits, but require extensive resources to implement. Because of this, it would be risky to choose to move forward with any of the options on their own. A few proposals, while having some notable benefits, would not aid in fulfilling strategic goals and would require extensive coordination to execute.

Some of the mid-tier options (such as advocacy groups and patient partnerships) scored well, particularly in sustainability and impact on accessibility, but not as well as the highest-scoring alternatives. These initiatives offer a balanced approach by combining resources from various sectors, suggesting short-term effectiveness and long-term potential. These initiatives are likely to be the most valuable when executed simultaneously with the highest scoring options.

The matrix reveals that Mail-in Testing Kits, Telemedicine Services, Pharmacy-Led Initiatives, Utilizing Local Libraries, and Online Portals are the most worthwhile initiatives. Each of these ideas excel in almost all categories, indicating they are not only feasible and cost-effective, but accessible as well. Each of these options are well-received by the community, scalable, relatively quick to implement, and sustainable. This suggests that these initiatives are well-rounded and likely to be successful in enhancing PGx accessibility. Their high scores in community acceptance also imply that they are initiatives that the public is likely to embrace, which is crucial for the long-term success of any healthcare program.

A key insight from the results is the need for accessible options, such as mail-in testing and telemedicine, which has been highlighted as a concern by underserved populations [30]. Both were repeatedly highlighted as effective solutions to the logistical challenges of healthcare in rural settings. These methods could overcome the physical limitations faced by patients, allowing them to receive PGx services without leaving their homes.

The use of mail-in testing kits can address some of the logistical challenges associated with rural healthcare. By enabling patients to complete PGx testing from the convenience of their homes, this approach eliminates [31]. Mail-in kits also support accessibility, allowing a broader segment of the rural population to benefit from PGx insights without disrupting the daily lives of patients and their caregivers by eliminating in-person visits.

Telemedicine is another tool that can make delivering PGx services in rural settings more accessible. This is especially valuable for continuity of care and specialist consultations. Stakeholders noted the importance of ongoing support, particularly in managing chronic conditions or coordinating PGx counseling sessions. Telemedicine addresses these needs by streamlining interactions between patients and healthcare providers, which is essential in rural regions with limited access to specialized care. Additionally, telemedicine allows PGx clinics to rural patients without the substantial costs associated with physical infrastructure expansion [32].

Pharmacy-led initiatives are another approach that can utilize existing resources to improve accessibility to specialized services by serving as educators and facilitators [33]. As familiar and accessible figures in rural healthcare, pharmacists are well-positioned to guide patients through the process of PGx [34]. By incorporating pharmacists into the PGx model, healthcare providers can feel strongly that their patients are receiving reliable information and hand-on support in regard to PGx and the testing process. This approach integrates routine healthcare touchpoints already present in rural areas into a plan to educate patients on the benefits of PGx, potentially leading to higher adoption of PGx testing.

The results also suggested that a key success factor for the success of any new PGx initiative will be the amount of direct engagement with healthcare partners. Engaging partners effectively will require targeted outreach that educates on patient benefits, workflow integration, and effective patient interactions. By demonstrating how PGx can enhance patient care without interfering with their current practices, outreach efforts can encourage healthcare providers to include PGx as part of their standard of care.

A comprehensive outreach strategy is critical for building awareness and promoting the adoption of PGx. Stakeholder conversations underscored the value of targeted educational materials that explain the proven statistical benefits of PGx and explain the practicalities of integrating it within their practices. An effective strategy could include participation in rural healthcare conferences, where key representatives can serve as keynote speakers or panelists to highlight the benefits of PGx and share case studies demonstrating its value. Further initiatives may include offering continuing medical education (CME) credits for training modules, virtual workshops, and webinars, which would help embed PGx into the professional development pathways of rural healthcare providers [35].

In this context, scalability will largely depend on leveraging existing infrastructure to minimize costs while maximizing outreach. Similarly to pharmacy-led initiatives, libraries can serve as community education centers by equipping them with materials that raise awareness and understanding of the benefits of personalized medicine [36].

Scaling PGx adoption will require a structured approach that moves beyond pilot programs toward full integration within the healthcare system. This will depend on leveraging existing healthcare touchpoints, expanding telemedicine infrastructure, and ensuring sustainable funding models that make testing affordable. In rural communities, aligning PGx implementation with primary care and pharmacy services will be critical in ensuring that the intervention reaches patients effectively. Standardized clinical guidelines will also be essential in promoting adoption, allowing healthcare professionals to follow a consistent approach in integrating PGx testing into patient care.

Long-term policy support will be instrumental in determining scalability. Governments and healthcare regulators must ensure that PGx services are covered through public health programs, reducing financial barriers for patients. Revising reimbursement structures to include PGx consultations as part of routine care could encourage provider engagement, making testing more accessible and financially sustainable. Further, collaboration with insurers and pharmaceutical companies could create additional funding opportunities, expanding accessibility while ensuring cost-efficiency.

Localized engagement will also be essential for long-term success. Community-based initiatives, such as pharmacist-led educational sessions and library-based health literacy programs, could bridge the knowledge gap between providers and patients. These efforts would promote trust in PGx and encourage higher adoption rates, particularly in communities where health literacy and access to specialty care remain significant barriers. Further, aligning PGx implementation with existing public health initiatives may increase its viability. By positioning PGx as a complementary tool within broader healthcare efforts, such as chronic disease management programs, its adoption could be seamlessly integrated into current health priorities.

A phased implementation model may be the most effective strategy for scaling PGx adoption. An initial deployment in high-need rural areas, followed by gradual expansion into broader healthcare networks, would allow for continuous refinement of service delivery and integration strategies. Pilot programs in select communities could provide valuable data on patient outcomes, healthcare provider engagement, and cost-effectiveness, informing subsequent phases of expansion.

Beyond improving medication personalization, PGx integration has the potential to reshape rural healthcare by enhancing patient autonomy, reducing healthcare costs associated with medication-related hospitalizations, and improving overall healthcare efficiency. By equipping providers with precision medicine tools, PGx could contribute to a more data-driven approach to patient care, strengthening rural healthcare networks while expanding patient access to specialized services.

### 4.1. Limitations

This study has several limitations that warrant acknowledgment. The first limitation is that the sample size for stakeholder interviews was relatively small, which may not capture the full diversity of perspectives across all rural healthcare settings. The second limitation is that the lack of quantitative data in the study restricts the analysis to primarily qualitative insights, which may not fully capture the scale of barriers or opportunities. The third limitation is that the study did not conduct long-term evaluations of the proposed implementation strategies, making it difficult to assess sustainability and real-world effectiveness. Future research should focus on expanding the sample size, incorporating quantitative metrics, and conducting longitudinal studies to validate the proposed recommendations.

### 4.2. Ethical Considerations

Privacy concerns are extremely common when it comes to technology, especially with regard to disclosing personal medical information. Healthcare innovation has led to the development of numerous pieces of software, in which patient data can be stored securely, without concern for a breach of privacy.

## 5. Conclusions

Ensuring the long-term success of PGx adoption in rural healthcare requires a focus on scalability, stakeholder engagement, continuous improvement, and education. Each of these elements is crucial to building a sustainable model that adapts to the unique needs of rural communities. This study’s findings highlight key insights into the challenges and opportunities for PGx adoption in rural areas and provided a foundation for a practical, targeted action plan.

Analysis into the healthcare landscape in rural Ontario found that in general, those residing in rural areas do not receive the same level of care as those living in urban centers. Individuals in these communities usually must travel to more populated regions in order to receive specialized services. While it is difficult to attract healthcare professionals to practice in rural areas, since COVID-19 there has been an increase in technology in the healthcare industry, which can be used to increase accessibility for personized services throughout the province.

To expand PGx to rural communities, strategic frameworks such as the Diamond-E, Porter’s 5 forces, and the Strategy Triangle are used to assess barriers and identify opportunities for long-term growth. A key consideration is economic feasibility, as initiatives must be affordable for both providers and patients. Currently, collecting samples in rural areas is costly due to limited mobile services, high transportation costs, and logistics. While mobile clinics and mail-in kits are options, both come with significant expenses.

Competitively, PGx faces limited alternatives but must stay ahead of technological advances and traditional medicine. The market for PGx is expanding, with a high growth rate projected through 2030. Entrants could benefit from partnerships to improve service quality, speed, and cost-effectiveness compared to existing mail-in kits and educational services.

A successful strategy requires alignment with the values of personalized medicine, ensuring that PGx is accessible and effective in rural areas. The approach should involve skilled interpretation of genetic data, clear communication, data privacy, and a focus on educating both healthcare providers and patients. With the right infrastructure, rural expansion could lead to reduced adverse drug reactions, improved treatment outcomes, and cost savings in healthcare.

Through conversations with healthcare providers, pharmacists, and patients, it became evident that a successful integration of PGx into rural healthcare would rely heavily on addressing two primary areas: logistical accessibility and healthcare partner awareness.

A key insight from the results is the need for accessible options, such as mail-in testing and telemedicine. Both were repeatedly highlighted as effective solutions to the logistical challenges of healthcare in rural settings. These methods could overcome the physical limitations faced by patients, allowing them to receive PGx services without leaving their homes.

Additionally, the results suggested that a key factor for the success of any new PGx initiative will be direct engagement with healthcare partners. Engaging partners effectively will require targeted outreach that educates on patient benefits, workflow integration, and effective patient interactions. By demonstrating how PGx can enhance patient care without adding complexity to existing practices, outreach efforts can encourage healthcare providers to include PGx as part of their standard of care.

Finally, using a decision matrix to rank various initiative options against key criteria allowed for the identification of the most worthwhile opportunities for expansion. Specifically, a mail-in-testing initiative would be attractive due to its relatively low cost, ability to increase accessibility, and its ability to satisfy a large increase in demand for PGx testing. This can be used in combination with telemedicine, allowing healthcare professionals to connect with patients in rural communities for initial consultations as well as to explain results, thus increasing timeliness. Finally, an outreach initiative on its own may not increase the accessibility of PGx testing, but used in combination with mail-in-testing and telemedicine it can aid in increasing awareness of the benefits, and thus the demand. If executed correctly, this trio of initiatives can allow for a seamless and successful integration of PGx into rural communities, not only increasing accessibility for specialized services but also improving the health of countless individuals.

## Figures and Tables

**Figure 1 jpm-15-00110-f001:**
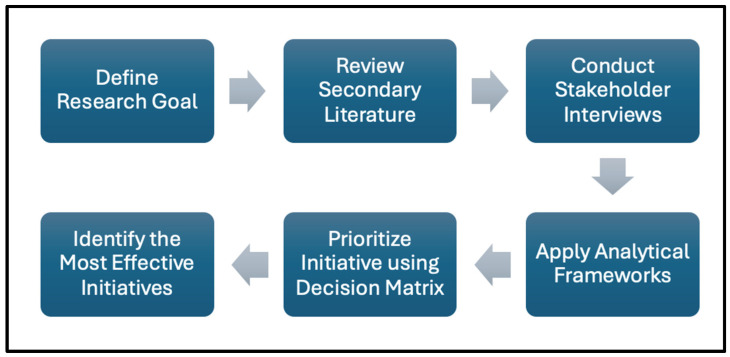
Research process flowchart.

**Figure 2 jpm-15-00110-f002:**
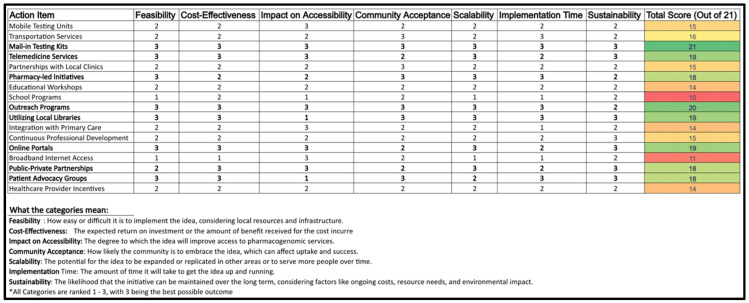
Decision matrix for pharmacogenomic initiatives.

## Data Availability

New data supporting reported results is unavailable due to privacy or ethical restrictions.

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
