# Peer review of "Enhancing Rural Healthcare Accessibility: A Model for Pharmacogenomics Adoption via an Outreach-Focused Integration Strategy"

_jpm, 2025, doi:10.3390/jpm15030110_

Round 1
Reviewer 1 Report
Comments and Suggestions for Authors
General
The manuscript addresses an important and timely topic in the field of personalized medicine. The proposed approaches are practical and methodologically sound, but the text is too long and contains repetitions that affect readability. In particular, the results and discussion could be made clearer and more concise.
Major
Introduction: The introduction is redundant in places and should be tightened. It should focus more on the relevance and novelty of the work without repeating general descriptions.
Results: This section contains excessive explanations and literature references that belong in the discussion. The results should be clearly structured and focused exclusively on the essential findings.
Discussion: The discussion repeats parts of the results and does not sufficiently address the long-term implications of the proposed measures. It should be made clearer how the approaches could be scaled and implemented.
Ethics and genetic counseling: The aspect of genetic counseling is not sufficiently addressed. It should be more clearly stated how ethical standards can be ensured and patients supported in the context of pharmacogenomics.
Minor
Methods: The detailed description of the frameworks (Diamond-E, Porter's Five Forces) is too extensive for the main text. These should be moved to the Supplementary Materials.
Appendices: Longer content such as interview guides or detailed framework explanations could also be moved to the Supplementary Materials to improve the readability of the main text.
Redundancies and conciseness: Terms such as "multi-facetted" and "innovative" are often overused in AI-generated or formal writing and should be avoided. General statements and repetitions should be removed to tighten up the text and present the core messages more clearly.
Author Response
Comment: The manuscript addresses an important and timely topic in the field of personalized medicine.
The proposed approaches are practical and methodologically sound, but the text is too long and contains repetitions that affect readability. In particular, the results and discussion could be made clearer and more concise.
Revision: The text has been restructured to enhance readability. Repetitions have been eliminated shortening the text. The results section especially has been reduced as mentioned in the comments below.
Comment: Introduction: The introduction is redundant in places and should be tightened. It should focus more on the relevance and novelty of the work without repeating general descriptions.
Revision: Condensed to reduce redundancies; Increased focus on the relevance and novelty of the work
Comment: Results: This section contains excessive explanations and literature references that belong in the discussion. The results should be clearly structured and focused exclusively on the essential findings.
Revision: Sections of the results that are explanatory in nature have been moved to the discussion section. Only necessary literature references as part of the literature review results are included in the results. The results are now focused on essential findings and redundancies have been eliminated.
Comment: Discussion: The discussion repeats parts of the results and does not sufficiently address the long-term implications of the proposed measures. It should be made clearer how the approaches could be scaled and implemented.
Revision: Long-term implications have been made clearer in lines 444-449. How it can be scaled is now mentioned in lines 421-443. As the results have been shortened, the discussion no longer contains repetition of things mentioned in the results.
Comment: Ethics and genetic counseling: The aspect of genetic counseling is not sufficiently addressed. It should be more clearly stated how ethical standards can be ensured and patients supported in the context of pharmacogenomics.
Revision: This section has been added in lines 461-465
Comment: Methods: The detailed description of the frameworks (Diamond-E, Porter's Five Forces) is too extensive for the main text. These should be moved to the Supplementary Materials.
Revision: Supplementary Materials has been created
Comment: Appendices: Longer content such as interview guides or detailed framework explanations could also be moved to the Supplementary Materials to improve the readability of the main text.
Revision: Supplementary Materials has been created
Comment: Redundancies and conciseness: Terms such as "multi-facetted" and "innovative" are often overused in AI-generated or formal writing and should be avoided. General statements and repetitions should be removed to tighten up the text and present the core messages more clearly. 
Revision: Text has been edited to reduce the use of such terms
Reviewer 2 Report
Comments and Suggestions for Authors
1. The authors stated that "A multi-method qualitative approach was employed, involving semi-structured interviews with key stakeholders and a qualitative review of secondary literature focused on rural healthcare and PGx." Please provide more details on the number of key informants included in the study and the criteria used for their selection.
2. Regarding the semi-structured interview tools, please clarify the steps involved in their development. This could include details on the design process, validation, and any pilot testing conducted.
3. For the qualitative review of secondary documents, please elaborate on the techniques and tools used for data analysis. Specify whether thematic analysis, content analysis, or any other method was applied.
4. Please justify the selection of the Diamond-E Framework and Porter’s Five Forces Framework in this study. Explain their relevance to your research context and how they contribute to the study's objectives.
5. Consider presenting your research process in a visual format, such as a flowchart or diagram, to enhance clarity and comprehension.
6. Please provide a diagram summarizing your complete research findings for better visualization and reader engagement.
Author Response
Comment: 1. The authors stated that "A multi-method qualitative approach was employed, involving semi-structured interviews with key stakeholders and a qualitative review of secondary literature focused on rural healthcare and PGx." Please provide more details on the number of key informants included in the study and the criteria used for their selection. 
Revision: Lines 80-92 have been updated to display the number of key stakeholders, the methods for how they were chosen, as well as clearer direction to find the sample breakdown in the appendix.
Comment: 2. Regarding the semi-structured interview tools, please clarify the steps involved in their development. This could include details on the design process, validation, and any pilot testing conducted. 
Revision: Lines 93-109 have been updated egarding clarity on the steps involved in the development of interview tools
Comment: 3. For the qualitative review of secondary documents, please elaborate on the techniques and tools used for data analysis. Specify whether thematic analysis, content analysis, or any other method was applied. 
Revision: Lines 111-114 now specify what type of analysis was used to review secondary documents and how it applies to our research
Comment: 4. Please justify the selection of the Diamond-E Framework and Porter’s Five Forces Framework in this study. Explain their relevance to your research context and how they contribute to the study's objectives. 
Revision: Lines 126-126 contain the justification of the relevance of the frameworks to our study
Comment: 5. Consider presenting your research process in a visual format, such as a flowchart or diagram, to enhance clarity and comprehension. 
Revision: A diagram has been added in line 77
Comment: 6. Please provide a diagram summarizing your complete research findings for better visualization and reader engagement. 
Revision: The decision matrix is how we have summarized our findings in terms of visualization.
Round 2
Reviewer 1 Report
Comments and Suggestions for Authors
The revisions align well with my comments.